# Antifungal Activity of Rhizosphere Bacillus Isolated from *Ziziphus jujuba* Against *Alternaria alternata*

**DOI:** 10.3390/microorganisms12112189

**Published:** 2024-10-30

**Authors:** Qiang Zou, Yumeng Zhang, Xinxiang Niu, Hongmei Yang, Min Chu, Ning Wang, Huifang Bao, Faqiang Zhan, Rong Yang, Kai Lou, Yingwu Shi

**Affiliations:** 1Institute of Microbiology, Xinjiang Academy of Agricultural Sciences, Urumqi 830091, China; m18827019154@163.com (Q.Z.); zhangyumeng4204@163.com (Y.Z.); yanghm@xaas.ac.cn (H.Y.); chumin@xaas.ac.cn (M.C.); wangning@xaas.ac.cn (N.W.); bbhf777@xaas.ac.cn (H.B.); zhanfaqiang@xaas.ac.cn (F.Z.); yangrong@xaas.ac.cn (R.Y.); loukai@tsinghua.org.cn (K.L.); 2College of Life and Science and Technology, Xinjiang University, Urumqi 830046, China; 3Institute of Soil, Fertilizer and Agricultural Water Conservation, Xinjiang Academy of Agricultural Sciences, Urumqi 830091, China; niuxx@xaas.ac.cn; 4Key Laboratory of Agricultural Environment in Northwest Oasis of Ministry of Agriculture and Countryside, Urumqi 830091, China; 5Xinjiang Laboratory of Special Environmental Microbiology, Urumqi 830091, China

**Keywords:** jujube fruit black spot disease, *Bacillus velezensis*, growth characteristics, control effect

## Abstract

The serious impact of *Alternaria alternata* on jujube black spot disease has seriously affected the quality and yield of jujube, constraining the sustainable development of the jujube industry. The purpose of this study was to isolate and screen highly effective biocontrol strains of jujube black spot disease from jujube rhizosphere soil. Thirty-three soil samples were collected from four regions in southern Xinjiang. The strains with antagonistic effects were isolated and screened by the dilution spread method and plate confrontation method and identified by morphological, physiological, and biochemical characteristics, as well as 16S rDNA, gyrB, and rpoB gene sequences. Indoor and field efficacy experiments were conducted to determine their biocontrol effect. A total of 110 strains with antibacterial activity were selected, and one strain, Bacillus velezensis 26-8, with a stable antagonistic effect was further tested. Biological characteristic experiments showed that strain 26-8 could grow at NaCl concentrations of 0.5–10% and pH 4.0–9.0. The biocontrol experiment results showed that Bacillus velezensis 26-8 could achieve an 89.83% control effect against black spot disease. In conclusion, strain 26-8 has good salt and alkali tolerance, exerts a good control effect on jujube black spot disease, and is worthy of further study.

## 1. Introduction

The jujube (*Ziziphus jujube* Mill) is a natural medicinal and edible plant that is rich in various nutrients, including vitamins, amino acids, and proteins [1,2]. The jujube industry serves as a key pillar for economic development in Xinjiang, China. However, in recent years, the emergence of jujube black spot disease has impeded the industry’s growth, resulting in significant economic losses for fruit farmers and adversely affecting the quality and yield of jujube fruits [3]. Currently, chemical control is the primary method for managing plant diseases; however, the long-term use of chemical pesticides can lead to environmental pollution and pose risks to human health. Consequently, biocontrol, as a new green and non-polluting control method, is gaining increasing attention and is expected to gradually replace chemical control [3,4,5].

Biocontrol bacteria are mainly distributed in the soil around the rhizosphere of plants and are an indispensable part of the microbial community structure in the rhizosphere soil. Biocontrol bacteria have the advantages of fast reproduction, simple fermentation, and stable control effect and are the main source of biocontrol agents. The biocontrol bacteria are mostly Bacillus, which is distributed in soil, air, and water. It can inhibit the reproduction of many pathogenic bacteria.

The spore-forming Bacillus is currently the most widely used biocontrol agent due to its rapid reproduction and strong resistance to environmental stressors. Among the various species of Bacillus, *Bacillus velezensis* has been extensively studied and has demonstrated effective disease prevention and growth-promoting properties [6]. For instance, Zhu Li et al. found that *Bacillus velezensis* SM905 powder exhibited an 80.74% efficacy in preventing iron tuber charcoal rot, outperforming the effects of carbendazim [7]. Similarly, Zhang Nuoni et al. reported that *Bacillus velezensis* ZF-10 achieved a relative prevention rate of 61.85% against the tobacco mosaic virus in pot trials [8]. Additionally, Li et al. discovered that *Bacillus velezensis* Ba-0321 provided an 81.00% prevention rate for tobacco root rot [9]. These studies suggest that *Bacillus velezensis* is a promising strain for biocontrol against various diseases and has significant application potential. However, there are currently limited reports on the effectiveness of *Bacillus velezensis* in controlling jujube black spot disease in field conditions, and relatively few products demonstrate consistent efficacy. The practical application of this biocontrol agent is influenced by numerous external factors, including soil conditions and geographical climate [6,10,11,12]. Therefore, further selection of strains with enhanced adaptability is crucial for expanding the resources available for biocontrol agents.

This study utilized the rhizosphere soil of Xinjiang jujube as the experimental material to isolate and screen Bacillus strains exhibiting antibacterial activity against *Alternaria alternata*. The strains were identified based on their morphological, physiological, and biochemical characteristics and molecular biology methods. Additionally, their growth characteristics were clarified, and the in vitro control effect of the strains on jujube black spot disease was investigated. This research provides valuable germplasm resources for the biological control of jujube black spot disease.

## 2. Materials and Methods

### 2.1. Isolation of Soil Microorganisms

A five-point sampling method was used to collect rhizosphere soil samples from jujube orchards in Shaya (47°17′14″ N, 82°42′30″ E), Wensu (41°18′27″ N, 80°34′17″ E), Moyu (37°9′45″ N, 79°38′0″ E), and Zepu (38°9′51″ N, 77°11′10″ E) of Xinjiang, China. We weighed 10.0 g of the soil sample, transferred it to a triangular flask containing 90 mL of sterile water and an appropriate amount of glass beads, shook it at a speed of 150 r/min for 20 min on a shaker to prepare soil suspension, diluted the soil suspension using a 10-fold dilution method to prepare concentrations of 10^−3^, 10^−4^, and 10^−5^, placed them in a constant temperature water bath at 80 °C for 15 min, evenly spread 200 μL of the soil dilution on NA agar plates, let it stand for 20 min, and then inverted the plates and incubated them at 30 °C for 48 h in a constant temperature incubator, with 3 replicates for each treatment. We then selected representative single colonies for purification and cultivation.

### 2.2. Antimicrobial Screening

We used the flat confrontation method to screen for antagonistic *Bacillus* spp. [13]. *Alternaria alternata* was isolated and preserved from the junction of diseased and healthy jujube fruits in this experiment, with the pathogen of jujube black spot as the target bacteria. We used a sterile hole punch to take 6 mm agar discs of *A*. *alternata*, placed them in the center of the PDA plate, spotted the antagonistic bacteria in four corners 2 cm away from the center with 3 parallel strains per strain, and incubated them at a constant temperature of 28 °C for 5 days. We then observed whether each strain had an antibacterial effect and recorded the strains with antibacterial activity.

We refer readers to the method described by Yang Di et al. [14] for the secondary screening of the initial strains. First, we inoculated 100 μL of *A. tenuissima* spore suspension onto the PDA medium and allowed it to stand for 10 min. Using a sterile hole punch, we created four equidistant holes, each 1.5 cm from the center of the PDA plate, following a cross pattern. We then added 100 μL of each antagonistic bacterial sterile filtrate to each hole and incubated it at 28 °C for 5 days, with three replicates per strain. We then observed the presence of inhibition zones and measured their diameters using the cross method.

### 2.3. Strain Identification

The methods for identifying antagonistic bacteria through morphological, physiological, and biochemical means are as follows: inoculate the purified antagonistic bacteria into nutrient agar (NA) medium and incubate at 30 °C for 48 h; observe the morphological characteristics of the colonies; and select colonies for Gram staining to examine the morphological features of the bacterial cells. The physiological and biochemical tests were performed according to the methods outlined in references [15,16,17,18].

The molecular biological identification methods for antagonistic bacteria are outlined as follows. First, we utilized a DNA extraction kit to isolate the genomic DNA of the antagonistic bacteria. Next, we used the extracted bacterial genomic DNA as a template to amplify target fragments with universal primers for the bacterial 16S rDNA, gyrB, and rpoB genes. The PCR amplification products should be analyzed using 1.7% agarose gel electrophoresis, and the clarity of the electrophoresis bands can be observed with a gel imaging system [19,20]. For bands that exhibited clear amplification products, we sent them to Shanghai Sangon Biotech Co., Ltd. (Shanghai, China) for sequencing. We then employed SeqMan software (version 12.0) to assemble the sequences and subsequently submitted them to the NCBI database for similarity comparison analysis using BLAST software 2.16.0. A phylogenetic tree can be constructed using the Neighbor-Joining method in MEGA7.0 software to determine the classification status of the strains. The amplification system and conditions are detailed in Table 1.

### 2.4. Determination of Antagonistic Bacterial Strain Growth Characteristics

The determination method of antagonistic bacteria growth curve is as follows: inoculate the antagonistic bacteria activated into 50 mL of the NB medium; shake the culture at 28 °C and 180 r/min for 24 h to obtain seed solution; inoculate the seed solution with 1% inoculum size into the NB medium; and shake the culture at 28 °C and 180 r/min, sampling every 2 h to measure the OD600 nm value using a spectrophotometer, with the sterile NB medium as the control and 3 replicates for each treatment.

The method for determining the effect of temperature on the growth of antagonistic bacteria is as follows: inoculate the antagonistic bacteria into the nutrient broth (NB) medium at a concentration of 1%; and incubate the culture at temperatures of 24 °C, 28 °C, 32 °C, 36 °C, and 40 °C, shaking at 180 rpm for 24 h. The viable bacterial count was assessed using plate-counting techniques.

The method for measuring the effect of pH on the growth of antagonistic bacteria is outlined as follows: prepare the nutrient broth (NB) culture medium at pH levels of 4, 5, 6, 7, and 8; incubate the cultures at 28 °C with shaking at 180 revolutions per minute (r/min) for 24 h; after incubation, determine the number of viable bacteria using the plate-counting method.

The method for determining the effect of NaCl concentration on the growth of antagonistic bacteria is as follows: prepare the nutrient broth (NB) culture medium with NaCl concentrations of 0.5%, 1%, 2%, 5%, 7%, and 10%; incubate the cultures at 28 °C and 180 rpm for 24 h; and assess the number of viable bacteria using the plate-counting method.

### 2.5. Evaluation of Strain Biocontrol Ability

The method for determining the effect of strain 26-8 on the spore germination of *Alternaria alternata* was conducted as follows: The concentration of the *A. alternata* spore suspension was adjusted to 1 × 10^6^ spores/mL using sterile water. The experiment consisted of the following treatments: a control group, which included 1 mL of the pathogenic spore suspension mixed with 1 mL of the nutrient broth (NB) culture medium; and a treatment group, where the pathogenic spore suspension was mixed with the supernatant of antagonistic bacteria fermentation at a 1:1 ratio. Each treatment was incubated at 28 °C for 24 h. Subsequently, 30 μL of the mixture was sampled for microscopic observation, and a total of 500 spores were examined to calculate the spore germination rate, with three replicates for each treatment. The spore germination inhibition rate (%) was calculated using the following formula: Spore germination inhibition rate (%) = [(Control spore germination rate − Treatment spore germination rate)/Control spore germination rate] × 100. The reason for this was improved clarity, technical accuracy, and readability by refining the sentence structure, enhancing vocabulary, and ensuring proper scientific notation.

The experimental methods for in vitro prevention of jujube fruit diseases are as follows: First, wash the jujube fruits with sterile water. Next, soak them in a 2% sodium hypochlorite solution for 5 min, and then rinse them three times with sterile water before allowing them to air dry naturally. Using a sterile needle, create a deep wound approximately 3 mm near the equator of each jujube fruit. Then, inoculate the wound with 10 μL of the *A. alternata* spore suspension and culture the fruit at 25 °C for 24 h. After this incubation period, inoculate with 10 μL of the antagonist solution. Using the NB culture medium as a control after inoculation, place the fruits in a plastic box, seal them with polyethylene film, and maintain a culture temperature of 25 °C with a relative humidity of 60%. Repeat this process three times, using 10 jujube fruits for each trial, and observe the diameter of the lesions on the jujube fruits every three days. Next, measure the lesion diameter using a cross method and calculate the average. The inhibition rate (%) is calculated using the following formula: Inhibition rate (%) = [(Control lesion diameter − Treated lesion diameter)/Control lesion diameter] × 100.

### 2.6. Statistical Analysis

Data were analyzed using standard analysis of variance (ANOVA) and Duncan’s multiple comparison tests with SPSS software 23. A *p*-value of less than 0.05 was considered statistically significant.

## 3. Results

### 3.1. Isolation and Screening of Strains

By employing the dilution spread method to isolate soil samples from four regions in Xinjiang, various morphologically distinct single colonies were selected for purification. The strains were initially screened using the plate confrontation method, resulting in the identification of 133 strains exhibiting antibacterial properties. Subsequently, the inhibition zone method was utilized for rescreening, leading to the selection of 110 strains with confirmed antibacterial activity. The antagonistic bacteria were categorized into five groups based on the diameter of the inhibition zones. Specifically, there were 14 strains with inhibition zone diameters ranging from 0 to 5 mm, 10 strains from 5 to 10 mm, 14 strains from 10 to 15 mm, 28 strains from 15 to 20 mm, and 44 strains from 20 to 30 mm (Table 2). Strain 26-8, which demonstrated relatively stable antibacterial activity, was selected, with an inhibition zone diameter of 25.37 ± 0.37 mm (Figure 1).

### 3.2. Identification of Antagonistic Strains

Strain 26-8 forms circular or irregular colonies on nutrient agar (NA) plates, exhibiting a rough, white, opaque surface. The bacteria are rod-shaped and Gram-positive (Figure 2). Physiological and biochemical tests indicated that strain 26-8 yielded positive results in the methyl red test, Voges–Proskauer (V-P) test, catalase test, oxidase test, nitrate reduction test, and hydrogen sulfide (H2S) gas production test. Additionally, strain 26-8 demonstrated the ability to hydrolyze starch, liquefy gelatin, ferment glucose, and utilize citrate, D-mannose, D-mannitol, and D-xylose, but it did not utilize L-arabinose (Table 3). Based on its morphological and physiological characteristics, along with the guidelines from the common bacterial identification manual, strain 26-8 was preliminarily identified as a *Bacillus* species.

Through the analysis of the 16S rDNA, gyrB, and rpoB gene sequences of strain 26-8, segments of approximately 1500 bp, 1200 bp, and 400–600 bp were amplified, respectively (Figure 3). The sequencing results were submitted to the NCBI database for BLAST comparison analysis, and gene accession numbers were obtained. The Neighbor-Joining method in MEGA 7.0 software was employed to construct a phylogenetic tree (Figure 4). The results indicated that all three gene sequences clustered together with *Bacillus velezensis*, exhibiting similarities of 99.28%, 99.64%, and 100%, respectively. In conjunction with morphological and physiological biochemical characteristics, strain 26-8 was identified as *Bacillus velezensis*.

### 3.3. Determination of Strain Growth Characteristics

From Figure 5, it is evident that *Bacillus velezensis* strain 26-8 exhibits slow growth from 0 to 4 h. Between 4 and 16 h, the strain experiences rapid growth, entering the logarithmic growth phase. After 18 h, the biomass of strain 26-8 stabilizes, with an optical density (OD600 nm) value of 1.7812. According to Figure 6, strain 26-8 can thrive within a temperature range of 24 °C to 40 °C, with a maximum viable count of 7.6 × 10^8^ CFU/mL observed at 28 °C. Figure 7 indicates that *B. velezensis* 26-8 can grow at a pH range of 4.0 to 9.0, with the highest viable count of 8.89 × 10^8^ CFU/mL recorded at pH 7. Growth of the strain is inhibited at pH levels below 5.0 or above 9.0. As illustrated in Figure 8, strain 26-8 can grow effectively in NaCl concentrations ranging from 0.5% to 5%, with no significant differences in growth observed within this range. However, at 7% and 10% salt concentrations, the viable cell count decreases, although the strain can still grow slowly under high-salinity conditions, yielding viable cell counts of 3.0 × 10^8^ CFU/mL and 6.23 × 10^4^ CFU/mL, respectively.

### 3.4. Effects of Strains on Spore Germination and Mycelial Growth of Alternaria alternata

The strain 26-8 fermentation broth exhibits a significant inhibitory effect on the germination of conidia from the pathogen responsible for jujube black spot disease (Figure 9), achieving an inhibition rate of 66.29% on conidial germination (Table 4). Following treatment with strain 26-8, the diameter of the fungal colonies of *Alternaria alternata* measured 39.71 mm, resulting in an inhibitory rate of 49.76% on hyphal growth (Table 5). These findings indicate that the sterile filtrate of strain 26-8 effectively inhibits the growth of *Alternaria alternata* hyphae (Figure 10).

### 3.5. Evaluation Results of Strain Biocontrol Ability

We pre-inoculated the chain spore suspension on jujube fruits and inoculated the antagonistic bacteria one day later. Observations were made every three days. It was noted that the lesion diameter in the treatment group was smaller than that in the control group. At 15 days, the lesion diameter in the treatment group measured 16.60 mm, significantly lower than the 31.70 mm observed in the control group. According to the results presented in Table 6, the average inhibition rate of black spot disease in jujube fruits for the treatment group was 50.47%. These results indicate that the antagonist 26-8 is effective in controlling black spot disease in jujube fruits.

## 4. Discussion

Jujube black spot disease is a fungal infection caused by *Alternaria alternata*, which is prevalent in southern Xinjiang and significantly impacts the development of the jujube industry [21]. Biological control has emerged as a prominent area of research in the prevention and management of plant diseases. Bacillus species are among the most widely utilized biocontrol microorganisms and offer considerable potential for application [22,23,24,25]. Within the Bacillus genus, interest in *Bacillus velezensis* as a biocontrol agent has increased in recent years. Studies have demonstrated that *B. velezensis* exhibits varying degrees of inhibitory effects against a range of plant diseases, including Fusarium [26,27,28,29], gray mold [30,31,32,33], and cotton Verticillium wilt [34,35]. The rapid and accurate identification of bacterial strains is essential for subsequent development and application. However, distinguishing Bacillus species solely based on phenotypic, physiological, and biochemical characteristics is often insufficient. With the rapid advancement of molecular biology techniques, molecular-level identification has become increasingly integrated into the process. The 16S rDNA gene sequence is widely utilized for the classification and identification of bacteria. For instance, Cui Lingxiao et al. successfully identified *Bacillus velezensis* strain 8-4 through 16S rDNA gene sequence analysis [36]. However, for closely related groups, 16S rDNA sequence analysis may fail, leading to inaccurate identification results [37]. In recent years, researchers have discovered that utilizing gene-encoded protein sequences, such as gyrA, gyrB, and rpoB, as molecular identifiers can address the limitations of 16S rDNA gene sequencing [19,38,39]. For instance, Feng et al. analyzed and identified the strain FY-C by integrating both 16S rDNA and gyrB gene sequences to enhance the accuracy of the identification results [40]. In this study, strain 26-8 was examined based on its morphological, physiological, and biochemical characteristics, as well as its 16S rDNA, gyrB, and rpoB gene sequences, leading to its identification as *Bacillus velezensis*. This research investigated the growth characteristics of *B. velezensis* 26-8, thereby laying the groundwork for the development and application of biocontrol strains. Strain 26-8 can thrive at temperatures ranging from 24 to 40 °C, with an optimal growth temperature of 28 °C. It can also tolerate pH levels between 4.0 and 9.0, with an optimal pH of 7.0 and salt concentrations from 0.5% to 10%. These findings are consistent with previous research on *Bacillus velezensis* [41,42,43]. However, they contrast with the study conducted by Yang Di et al., which identified the optimal pH for *Bacillus velezensis* as 5.5 [14]. Given the extensive land area in our country, characterized by diverse soil types and geographical environments, the growth and reproduction of microorganisms are influenced, leading to variations in their growth characteristics. Consequently, the identification of strains with enhanced environmental adaptability is crucial for biocontrol research. *B. velezensis* 26-8 exhibits strong salt and alkali tolerance, highlighting its significant biocontrol potential and establishing it as a valuable microbial resource for biocontrol applications.

Research has demonstrated that *Bacillus velezensis* can produce a variety of antibacterial metabolites, including lipopeptides, proteins, and polyketide compounds. Two endophytic antagonistic bacteria, *B. velezensis* ZJJDZY and ZJJDYB, exhibited significant antagonistic effects against various pathogenic fungi responsible for black spot disease, indicating their potential as biocontrol resources [44]. The control efficacy of *B. velezensis* GUAL210 against rose black spot reached as high as 60.96%, suggesting that GUAL210 holds promising prospects for application and development, and may serve as a viable alternative to chemical control agents [43]. This study found that the sterile filtrate of strain 26-8 can inhibit the growth of *Alternaria alternata*. It is hypothesized that one of the mechanisms by which strain 26-8 exerts its biocontrol effect is through the production of specific active substances that inhibit the growth of the pathogen; however, further research is necessary to identify the specific antibacterial substances and elucidate the biocontrol mechanisms involved. A strain of *B. velezensis* isolated during this experiment demonstrated a significant inhibitory effect on jujube black spot disease, exhibiting an inhibition zone diameter of 25.37 mm in the plate confrontation test and a control efficacy of 50.47% in the detached fruit treatment experiment. The control efficacy observed in this study differs from that reported in other research, such as the study by Song et al., which indicated that when *Bacillus amyloliquefaciens* K5-1 was inoculated first on jujube fruits, the control efficacy against winter jujube black spot disease reached 78.50% [45]. This discrepancy may be attributed to the experimental design, where the pathogen was inoculated first, followed by the antagonist. This sequence allowed the pathogen to occupy a favorable niche initially, resulting in a lower control efficacy compared to scenarios where the antagonist is inoculated first. Research on biocontrol agents should extend beyond laboratory conditions to ensure practical applicability. Field efficacy experiments are essential to validate and stabilize the effectiveness of these agents in real-world settings before further development as biological control agents.

## 5. Conclusions

In this study, 110 strains exhibiting antibacterial activity were screened, and the inhibition zone method was employed for further evaluation. Ultimately, a strain of *Bacillus velezensis*, designated as 26-8, was identified for its stable antagonistic effects. The plate confrontation experiment revealed that the inhibition zone diameter of *Bacillus velezensis* 26-8 against *Alternaria alternata* was 25.37 mm. The results from in vitro control experiments indicated that *Bacillus velezensis* 26-8 achieved a control efficacy of 50.47% against jujube black spot. Strain 26-8 demonstrated the ability to grow in the presence of 0.5% to 10% NaCl and within a pH range of 4.0 to 9.0. The control efficacy of *Bacillus velezensis* 26-8 against black spot disease reached an impressive 89.83%. Strain 26-8 exhibits strong saline-alkali tolerance and effective control over jujube black spot, highlighting its potential for further in-depth study.

## Figures and Tables

**Figure 1 microorganisms-12-02189-f001:**
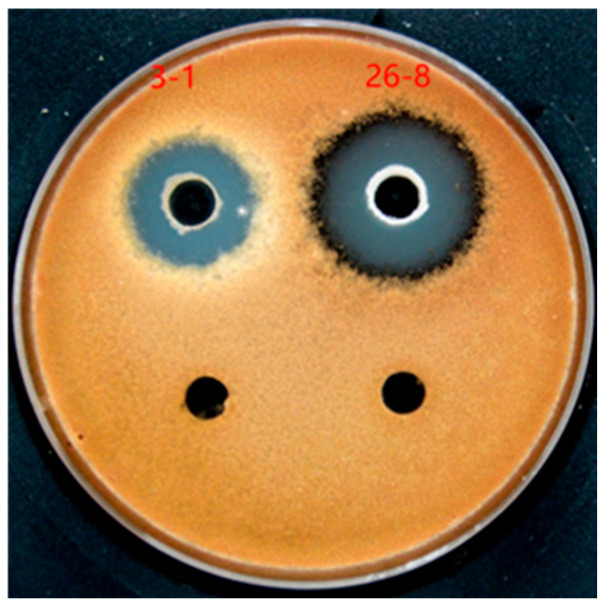
Inhibitory effect of antagonistic bacteria on Alternaria.

**Figure 2 microorganisms-12-02189-f002:**
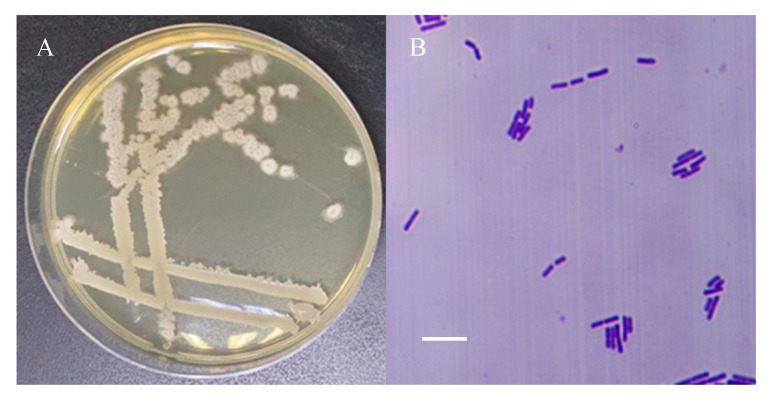
Colony morphology (**A**) and Gram staining (under 10 × 100 times optical microscope) (**B**) of strain 26-8. Bar = 5 μm.

**Figure 3 microorganisms-12-02189-f003:**
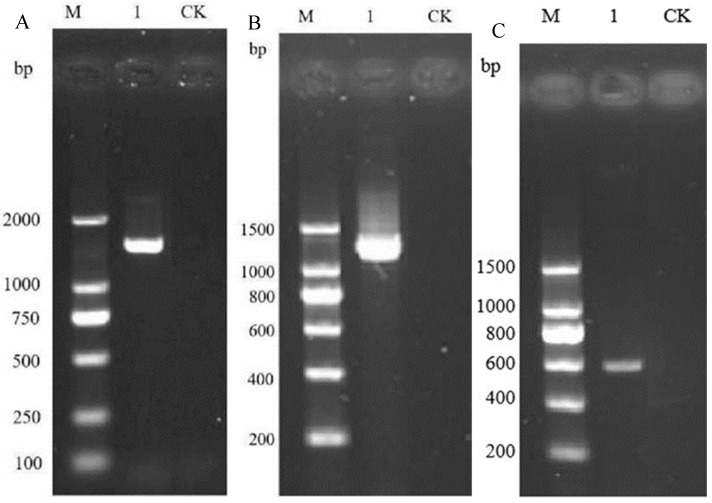
PCR gel electrophoresis based on 16S rDNA (**A**), gyrB (**B**), and rpoB (**C**) gene sequence.

**Figure 4 microorganisms-12-02189-f004:**
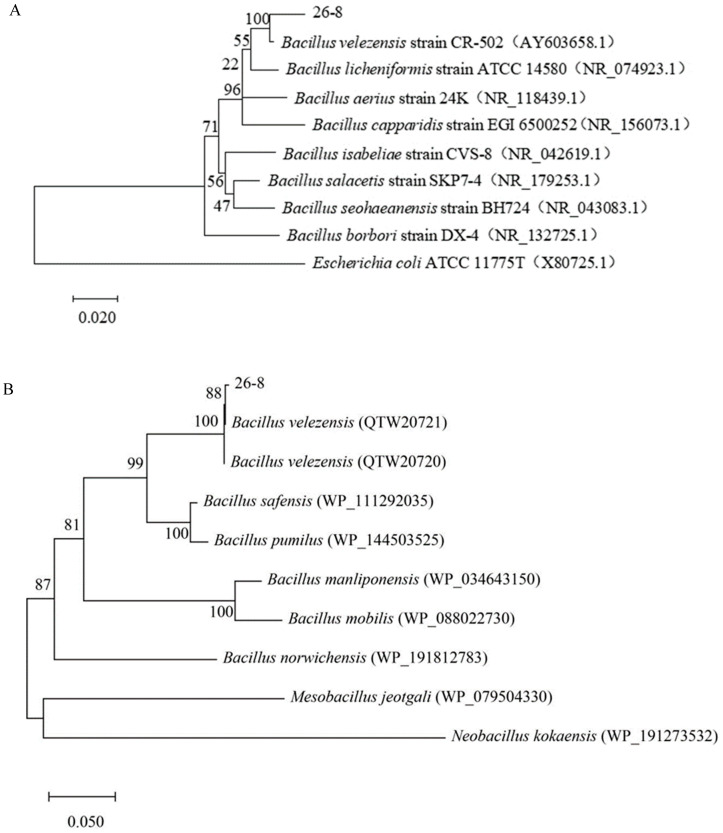
Phylogenetic tree based on 16S rDNA (**A**), gyrB (**B**), rpoB and (**C**) gene sequence. Branch termini are tagged in light of isolate species and GenBank accession numbers. The numbers above (or below) the nodes show the bootstrap values (50%) occurring after 1000 replications. Scale bars represent the average number of nucleotide substitutions per site.

**Figure 5 microorganisms-12-02189-f005:**
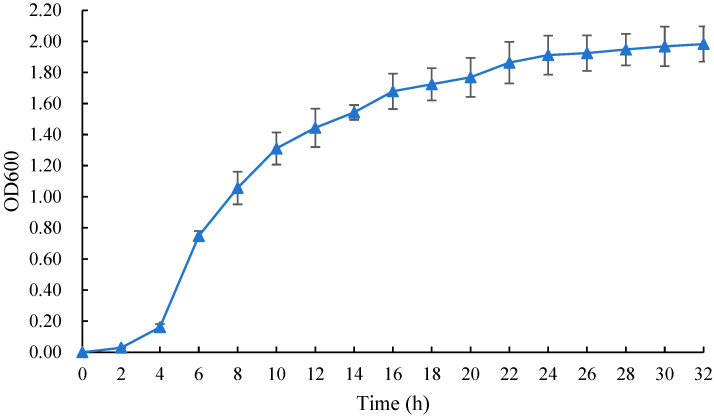
Growth curve of strain 26-8.

**Figure 6 microorganisms-12-02189-f006:**
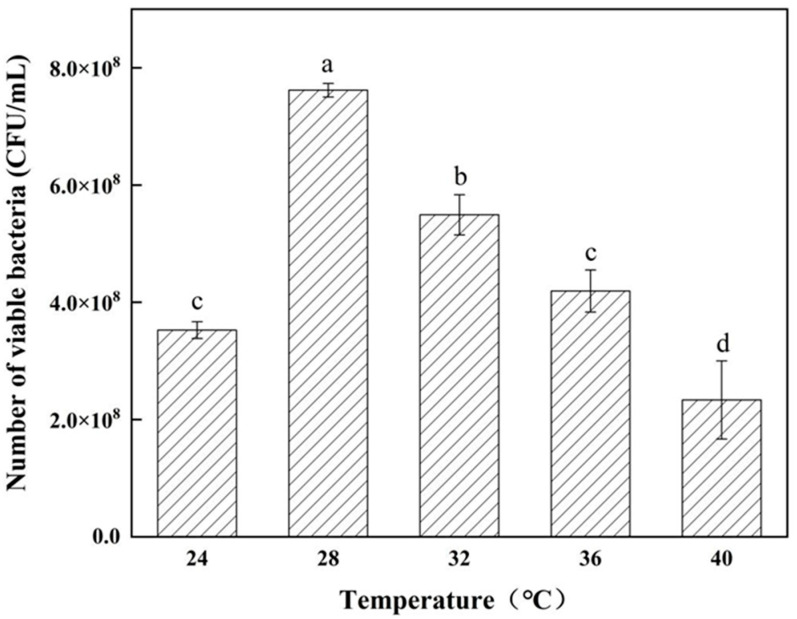
Effect of temperature on the growth of strain 26-8. Different lowercase letters represent significant differences between treatments (*p* < 0.05). Data marked with the same letter on the columns indicate no significant difference (*p* ≥ 0.05) according to Duncan’s multiple comparison tests.

**Figure 7 microorganisms-12-02189-f007:**
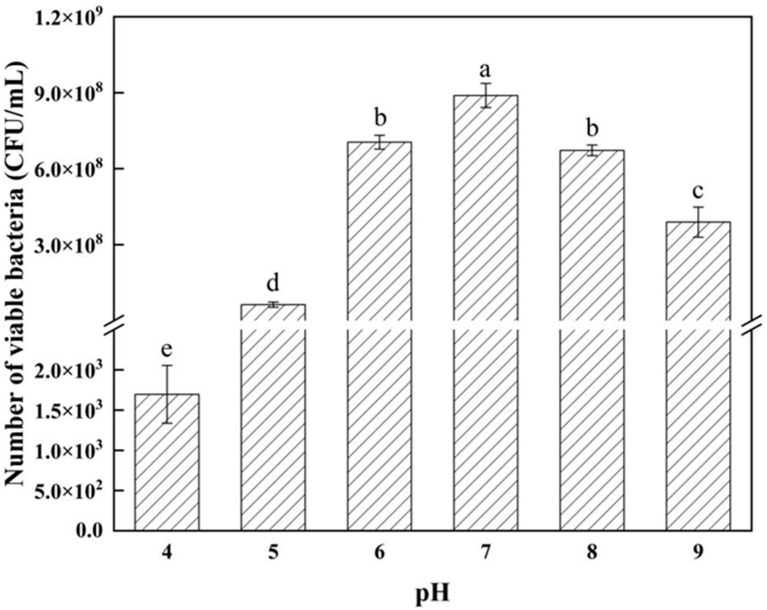
Effect of pH on the growth of strain 26-8. Different lowercase letters represent significant differences between treatments (*p* < 0.05). Data marked with the same letter on the columns indicate no significant difference (*p* ≥ 0.05) according to Duncan’s multiple comparison tests.

**Figure 8 microorganisms-12-02189-f008:**
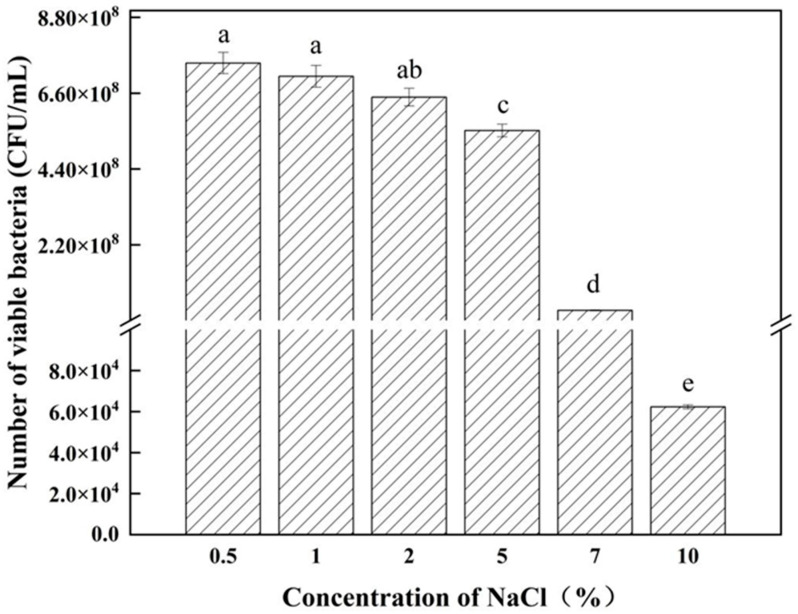
Effect of NaCl concentration on the growth of strain 26-8. Different lowercase letters represent significant differences between treatments (*p* < 0.05). Data marked with the same letter on the columns indicate no significant difference (*p* ≥ 0.05) according to Duncan’s multiple comparison tests.

**Figure 9 microorganisms-12-02189-f009:**
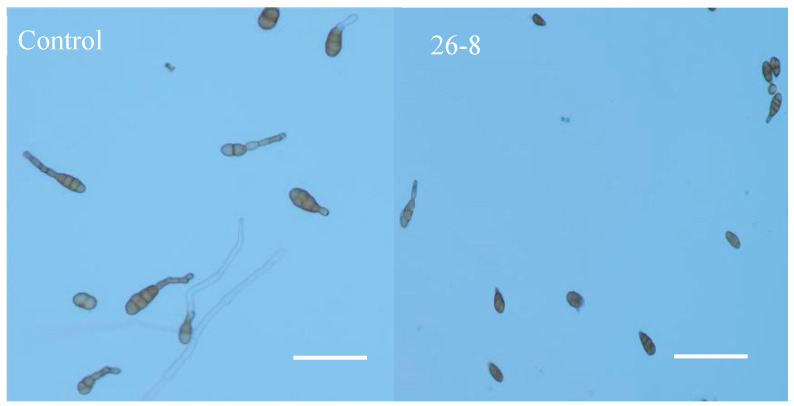
Microscopy of strain26-8 inhibition of conidium germination of *Alternaria alternata*. (under 10 × 40 times optical microscope). Scale bars: control, 26-8 = 10 μm.

**Figure 10 microorganisms-12-02189-f010:**
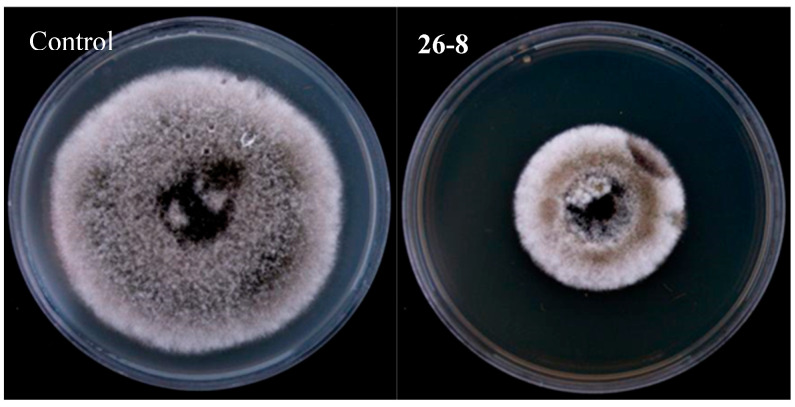
Diagram of inhibitory effect of antagonists on hyphal growth of *Alternaria alternata*.

**Table 1 microorganisms-12-02189-t001:** PCR amplification system and conditions.

Gene Segment	Primer	Reaction System	Reaction Conditions
16S rDNA	27F(5′-AGAGTTTGATCCTGGCTCAG-3′) 1492R(5′-GGTTACCTTGTTACGACTT-3′)	Mix 12.5 μL primer 0.5 μL template 1.0 μL ddH_2_O 11.0 μL	95 °C pre-denaturation 5 min. 95 °C degeneration 30 s, 58 °C annealing 15 s, 72 °C elongation 2 min, 20 cycles. 72 °C elongation 10 min.
gyrB	UP-1(5′-GAAGTCATCATGACCGTTCTGCAYGCNGGNGGNAARTTYGA-3′) UP-2r(5′-AGCAGGGTACGGATGTGCGAGCCRTCNACRTCNGCRTCNGTCAT-3′)		95 °C pre-denaturation 5 min. 94 °C degeneration 1 min, 54 °C annealing 1 min, 72 °C elongation 2 min, 25 cycles. 72 °C elongation 10 min.
rpoB	f(5′-AGGTCAACTAGTTCAGTATGGAC-3′) r(5′-AAGAACCGTAACCGGCAACTT-3′)		94 °C pre-denaturation 4 min. 94 °C degeneration 1 min, 51 °C annealing 1 min, 72 °C elongation 1 min, 25 cycles. 72 °C elongation 10 min.

**Table 2 microorganisms-12-02189-t002:** Inhibitory activity of 110 antagonistic strains against Alternaria.

Antifungal Activity	Inhibition Diameter (mm)	Number of Strains
-	0–5	14
+	5–10	10
++	10–15	14
+++	15–20	28
++++	20–30	44

Note: ++++: Very strong; +++: Strong; ++: Middle; +: Weak; -: Very weak.

**Table 3 microorganisms-12-02189-t003:** Physiological and biochemical characteristics of strain 26-8.

Items	26-8
Methyl red test	+
V-P text	+
Catalase test	+
Oxidse test	+
Amylolysis	+
Nitrate reduction	+
H2S production	+
Citrate solution test	+
Gelatin liquefaction	+
D-mannose	+
D-mannitol	+
D-xylose	+
D-arabinose	−
Oxidative fermentation of glucose	fermentation

Note: +: Positive; −: Negative.

**Table 4 microorganisms-12-02189-t004:** Inhibitory effect of strain 26-8 on spore germination of *Alternaria alternata*.

Treatment	Germination Rate (%)	Inhibition Rate (%)
CK	88.46 ± 2.05 ^a^	-
26-8	29.82 ± 0.85 ^b^	66.29

Data with the different lowercases letters (a, b) in same column indicated significantly different (*p* < 0.05, Duncan’s multiple range test).

**Table 5 microorganisms-12-02189-t005:** Inhibitory effect of strain 26-8 on hyphal growth of *Alternaria alternata*.

Treatment	Colony Diameter (mm ± SE)	Inhibition Rate (%)
CK	79.04 ± 0.62 ^a^	-
26-8	39.71 ± 0.31 ^b^	49.76

Data with the different lowercases letters (a, b) in same column indicated significantly different (*p* < 0.05, Duncan’s multiple range test).

**Table 6 microorganisms-12-02189-t006:** Effect of strain 26-8 on control of black spot of jujube fruit.

Time (d)	Spot Diameter (% ± SE) ^a^		Inhibition Rates (%)
	Treatment Group	Control Group	
3	5.47 ± 0.33 a	10.95 ± 0.77 a	50.01
6	8.39 ± 0.54 a	17.07 ± 1.55 b	50.88
9	10.46 ± 1.04 a	21.33 ± 0.61 b	50.95
12	14.28 ± 1.69 a	31.00 ± 2.61 b	53.95
15	16.60 ± 2.56 a	31.70 ± 0.86 b	46.57

^a^ Means within a column followed by a common lower-case letter are not significantly different (*p* > 0.05, Duncan’s multiple range test).

## Data Availability

Data are available upon request due to restrictions.

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
