# Peer review of "Antifungal Activity of Rhizosphere Bacillus Isolated from Ziziphus jujuba Against Alternaria alternata"

_microorganisms, 2024, doi:10.3390/microorganisms12112189_

Round 1
Reviewer 1 Report
Comments and Suggestions for Authors
Abstract
Please comply with MDPI guidelines, and maintain the abstract in less than 200 words
INTRODUCTION
Bacillus velezensis is not a "branch", it's just a species of the genera. please modify the statement
METHODS
Please improve the readability of the 2.1 section
please, write in "third person" the methods section
RESULTS
Please improve the quality of figures 6 to 8
figure 3 gives no extra information, it can be removed from the main text
please add letter to differentiate the 3 images on fig 4 (a, b , and c)
figure 9 and 10 can be fused on one single figure, since table 6 shows the data graphically presented on figure 10
DISCUSSION
Please add a table which compares at least 10 strains with Alternaria alternata antagonist effect. which compares the spot diameter and other critical variables
at the end of this section please discuss on the possible opportunities on the usage of this new strain, and what can be done to improve its inhibition activity
Author Response
A List of Responses to the Comments
Reviewer #1:
EVALUATION
Abstract
- Please comply with MDPI guidelines, and maintain the abstract in less than 200 words
Response: Thank you. We have carefully revised this abstract of the manuscript.
INTRODUCTION
- Bacillus velezensis is not a "branch", it's just a species of the genera. please modify the statement
Response: Thank you. We have carefully revised this sentence of the manuscript.
METHODS
- Please improve the readability of the 2.1 section
please, write in "third person" the methods section
Response: Thank you. We have carefully revised this section of the manuscript.
RESULTS
- Please improve the quality of figures 6 to 8
Response: Thank you. We have carefully improved these figures 6 to 8 of the manuscript.
- figure 3 gives no extra information, it can be removed from the main text
Response: Thank you. The electrophoresis bands gave the correctness of the amplified fragments and should be retained.
- please add letter to differentiate the 3 images on fig 4 (a, b, and c)
Response: Thank you. Different graphs in Fig.3 are marked with a, b, c.
- figure 9 and 10 can be fused on one single figure, since table 6 shows the data graphically presented on figure 10
Response: Thank you. We have carefully revised this figure 9 and 10 of the manuscript.
DISCUSSION
- Please add a table which compares at least 10 strains with Alternaria alternata antagonist effect. which compares the spot diameter and other critical variables.
Response: Thank you. We have added a table which compares at least 10 strains with Alternaria alternata antagonist effect. which compares the spot diameter and other critical variables.
- at the end of this section please discuss on the possible opportunities on the usage of this new strain, and what can be done to improve its inhibition activity
Response: Thank you. We have carefully revised this section of the manuscript.
Reviewer 2
In a paper entitled 'Antifungal Activity of Rhizosphere Bacillus Isolated from Ziziphus jujuba against Alternaria alternata', the authors focused on the isolation and characterisation of bacterial species with biocontrol potential against the pathogenic strain Alternaria alternata. In their work, the authors address the important issue of finding alternative methods to control fungal diseases that cause high crop losses.
Response: Thank you. Thank the reviewers for their positive evaluation of our research work.
The following is a review of the paper:
General comments:
- The paper needs linguistic correction as many sentences or phrases seem unnecessarily convoluted or difficult to understand.
Response: Thank you. We have made grammatical corrections to the paper.
Introduction:
- In the introduction, the authors should add some information on examples of bacteria-based biopreparations already in use. In particular, species belonging to the genus Bacillius. This is particularly relevant in the context of using their results in cultivated crops. In addition, when investigating the different properties of the isolate obtained, it is useful to include in the introduction information on the biocontrol mechanisms involved in the performance of the isolate under study.
Response: Thank you. We have carefully revised this section of the manuscript.
- In the aim of the paper, the authors write about Rhizoctonia solani and the rest of the paper is about Alternaria alternata. Is this a mistake?
Response: Thank you. We have carefully revised this sentence of the manuscript. Should be Alternaria alternata.
Materials and methods:
- Chapter 2.1
Coordinates of the sites from which soil samples were taken would be useful.
Response: Thank you. We 've added latitude and longitude.
2.2
- In the title of the paper, the authors write that the species obtained show antagonism against the fungus Alternaria alternata. And in this chapter the authors write that the experiments were also carried out against A. tenuissima. Was that the premise of the experiment? Has a mistake crept into this chapter?
Response: Thank you. This chapter is correct.
Using the flat confrontation method to screen for antagonistic Bacillus spp. [14]. Al-ternaria alternata was isolated and preserved from the junction of diseased and healthy ju-jube fruits by this experiment with the pathogen of jujube black spot as the target bacteria. Use a sterile hole punch to take 6 mm agar discs of A. alternata, place them in the center of the PDA plate, spot the antagonistic bacteria at four corners 2 cm away from the center, with 3 parallel strains per strain, and incubate at a constant temperature of 28 °C for 5 days. Observe whether each strain has an antibacterial effect, and record the strains with antibacterial activity.
2.4
- Did the authors determine the OD600 of the suspension with which they inoculated further experiments? This is very important information for further cultures where the authors determined the OD600 of the cultures in further experiments.
Response: Thank you. Fig. 6 is the result of this part.
- Table 1. needs to be checked and corrected as there are many blank spaces.
Response: Thank you. We have carefully revised this Table 1 of the manuscript.
Results.
- In Figure 4. the authors could colour code the location of their isolate.
Response: Thank you. We have carefully revised this sentence of the manuscript.
Fig. 4 Phylogenetic tree based on 16S rDNA(A)、gyrB(B)、rpoB(C) gene sequence.
Branch termini are tagged in light of isolate species and GenBank accession numbers. The numbers above (or below) the nodes show the bootstrap values (50%) occurring after 1000 replications. Scale bars represent the average number of nucleotide substitutions per site.
Chapter 3.3
- The authors describe changes in the bacterial biomass. By measuring the OD600, we cannot speak of a change in biomass, only an increase in the number of cells based on the optical density or by converting according to an appropriate scale to the number of CFU. To determine biomass, the bacteria obtained would have to be dried and weighed. I suggest that the authors modify this determination.
Response: Thank you. Because the growth curve to continuous measurement number, so with OD600 better solve the error caused by different sampling.
- In this subsection, the authors also need to work on the formatting. They write CFU once with a superscript and once with a ^ sign.
Response: Thank you. We have carefully revised this symbol of the manuscript.
3.4
- Wrong formatting in the caption of figure 9
Response: Thank you. We have carefully revised this sentence of the manuscript.
3.5
- The authors need to correct the text in this subsection. The term "control group control" sounds strange. Also, the authors write the word once in capital letters (CONTROL) and once in lower case (Control).
Response: Thank you. We have carefully revised this sentence of the manuscript.
Discussion
- I would avoid terms such as Hot topic
Response: Thank you. We have carefully revised this sentence of the manuscript.

Reviewer 2 Report
Comments and Suggestions for Authors
In a paper entitled 'Antifungal Activity of Rhizosphere Bacillus Isolated from Ziziphus jujuba against Alternaria alternata', the authors focused on the isolation and characterisation of bacterial species with biocontrol potential against the pathogenic strain Alternaria alternata. In their work, the authors address the important issue of finding alternative methods to control fungal diseases that cause high crop losses.
The following is a review of the paper:
General comments:
The paper needs linguistic correction as many sentences or phrases seem unnecessarily convoluted or difficult to understand.
Introduction:
In the introduction, the authors should add some information on examples of bacteria-based biopreparations already in use. In particular, species belonging to the genus Bacillius. This is particularly relevant in the context of using their results in cultivated crops. In addition, when investigating the different properties of the isolate obtained, it is useful to include in the introduction information on the biocontrol mechanisms involved in the performance of the isolate under study.
In the aim of the paper, the authors write about Rhizoctonia solani and the rest of the paper is about Alternaria alternata. Is this a mistake?
Materials and methods:
Chapter 2.1
Coordinates of the sites from which soil samples were taken would be useful.
2.2
In the title of the paper, the authors write that the species obtained show antagonism against the fungus Alternaria alternata. And in this chapter the authors write that the experiments were also carried out against A. tenuissima. Was that the premise of the experiment? Has a mistake crept into this chapter?
2.4
Did the authors determine the OD600 of the suspension with which they inoculated further experiments? This is very important information for further cultures where the authors determined the OD600 of the cultures in further experiments.
Table 1. needs to be checked and corrected as there are many blank spaces.
Results.
In Figure 4. the authors could colour code the location of their isolate.
Chapter 3.3
The authors describe changes in the bacterial biomass. By measuring the OD600, we cannot speak of a change in biomass, only an increase in the number of cells based on the optical density or by converting according to an appropriate scale to the number of CFU. To determine biomass, the bacteria obtained would have to be dried and weighed. I suggest that the authors modify this determination.
In this subsection, the authors also need to work on the formatting. They write CFU once with a superscript and once with a ^ sign.
3.4
Wrong formatting in the caption of figure 9
3.5
The authors need to correct the text in this subsection. The term "control group control" sounds strange. Also, the authors write the word once in capital letters (CONTROL) and once in lower case (Control).
Discussion
I would avoid terms such as Hot topic
Author Response

(The authors gave the same response as above.)

Reviewer 3 Report
Comments and Suggestions for Authors
Dear authors, the manuscript "Antifungal Activity of Rhizosphere Bacillus Isolated from
Ziziphus jujuba against Alternaria alternata" present a good research idea in the topic of biocontrol based on bacterial activity.
There are some changes that can be made to improve the current form of the manuscript.
Abstract - for this section, the aim and the materials and methods occupy most of the text. Make these parts more condensed and replace them with information related to the main findings of the research.
The Introduction section should be expanded, with the addition of more information related to the background of the research. The last paragraph, which contain the aim of the research, should be separated in smaller sentences, to provide the aim and the objectives/hypotheses of this study.
Materials and methods section - the authors should modify the sentences that sound like laboratory guidance. E.g. "Select representative single colonies for purification and cultivation.", "Using the flat confrontation method to screen for antagonistic Bacillus spp. [14].", "Use a sterile hole punch to take 6 mm agar discs...", "Refer to the method of Yang Di et al. [15] for the secondary screening of the initial strains. Inoculate 100μL of A. tenuissima spore suspension on PDA medium and let stand for 10 minutes."
Overall, this section provide all the necessary details for replication of the experiment. English language should be checked for this section.
Results section is explicit, presenting the findings of this research.
Discussion section - this section links the findings of the authors with other international references in the field.
Conclusion section - this section need to be expanded, with additional findings.
General comment - Avoid sentences longer than 4 rows. It is better to split them in shorter and concise sentences.
Comments on the Quality of English LanguageEnglish need to be checked.
Author Response

(The authors gave the same response as above.)

Round 2
Reviewer 3 Report
Comments and Suggestions for Authors
The authors have improved their work